# Top2b-Regulated Genes and Pathways Linked to Retinal Homeostasis and Degeneration

**DOI:** 10.3390/cells14120887

**Published:** 2025-06-12

**Authors:** Merna M. Ibrahim, Li Cai

**Affiliations:** Department of Biomedical Engineering, Rutgers University, Piscataway, NJ 08854, USA; mmi42@gsbs.rutgers.edu

**Keywords:** retinal homeostasis, topoisomerase II beta (Top2b), photoreceptor cells, transcriptional regulation, pathway analysis, gene expression, RNA sequencing, ciliary function, visual cycle

## Abstract

Retinal homeostasis and degeneration are significant contributors to global vision loss, with retinal health primarily assessed by the count and function of photoreceptor cells, the most abundant cells in the retina. Genomic studies have identified topoisomerase II beta (Top2b), an enzyme that untangles DNA supercoils to facilitate gene expression, as a critical transcriptional regulator for retinal health. This review aims to uncover and categorize genes linked to Top2b that are dynamically expressed during retinal degeneration, revealing shared and overlooked regulatory pathways. RNA sequencing data from wild-type and Top2b knockout mice revealed thousands of differentially expressed genes regulated by Top2b. By cross-referencing these genes with retinal degeneration datasets, including RetNet and the Gene Ontology Browser, we identified 44 Top2b-linked genes associated with retinal degeneration. These genes were grouped into fourteen functional categories: ciliary function and trafficking, metabolism, synaptic transmission, transcription factors and regulation, visual cycle, retinoids, and more. Key genes such as Bbs7, Ubb, Rbp4, Cetn2, Pik3r1, and Crx were explored, and their critical pathways for retinal health were outlined. This comprehensive catalog of 44 Top2b-linked retinal homeostatic genes will serve as a valuable resource for researchers. It provides new insights into the regulatory mechanisms underlying retinal homeostasis, setting the framework for the development of targeted therapeutic approaches and early intervention strategies for preventing photoreceptor loss.

## 1. Introduction

Progressive retinal degeneration (RD) is a major driver of visual impairment globally, with cases projected to rise 55% by 2050 [1]. Recent studies further highlight the growing concern, with a 2022 JAMA Ophthalmology study reporting that in 2019, approximately 19.8 million Americans aged 40 and older were living with age-related macular degeneration (AMD), including 1.49 million with vision-threatening AMD [2]. Globally, projections estimate the number of people with AMD will increase from 196 million in 2020 to 288 million by 2040 [3]. This underscores the substantial and growing burden of retinal diseases worldwide.

Encompassing many debilitating eye disorders, retinal degeneration is often characterized by the progressive loss of photoreceptor cells in the retina [4]. As these once-abundant retinal cells begin to degrade, damaged vision soon follows. This description, a somewhat mosaic label, comprises conditions such as retinitis pigmentosa (RP), AMD, and more [5]. It is worth noting that these are all mechanistically distinct disorders. Inherited retinal dystrophies (IRDs) typically result from monogenic mutations affecting specific cellular processes, while age-related macular degeneration (AMD) represents a complex, multifactorial disease influenced by aging, oxidative stress, and inflammation. Our analysis examines which Top2b-regulated genes are associated with these distinct disease categories. These not-well-understood disorders significantly affect millions of individuals worldwide, result in heavy socioeconomic burdens, and profoundly impact an individual’s quality of life.

The retina converts light signals into neural impulses [5], with photoreceptor cells vital in this process. Photoreceptor cells in the outer nuclear layer (ONL) face unique challenges: high metabolic demands, constant oxidative stress from light exposure, and the need for precise protein trafficking through the connecting cilium [6]. These specialized neurons require extensive transcriptional programs to maintain their complex structure and function, making them particularly vulnerable to disruptions in gene expression regulation [7]. This vulnerability underlies both inherited retinal dystrophies (IRDs) and contributes to age-related macular degeneration (AMD), though through distinct mechanisms.

Retinal homeostasis research assists scientists in deepening their understanding of neurodegeneration and tissue regeneration fundamentals. Due to the retina’s accessibility and thoroughly outlined structure, studying retina homeostasis is a promising model for studying the genetic and cellular mechanisms powering complex disorders. Innovations in gene sequencing technology and molecular biology have identified hundreds of retinal degeneration-associated genes, allowing researchers to gain insights into disease pathology and potential therapeutic targets [7].

The critical retinal gene region of DNA is managed by the enzyme topoisomerase IIb (Top2b). The enzyme unwinds DNA supercoiling by detangling knots that arise during the transcription of long genes. Long genes are especially relevant for the retina since they often encode proteins critical for photoreceptor and general retinal survival [8]. This is all achieved by Top2b’s double-strand break formations in DNA, offering much-needed relief from torsional stress. This relief upregulates transcriptional machinery access to these crucial long genes. When the Top2b function goes awry, transcription delays and repercussions arise. The subsequent supercoiling due to this dysfunction increases the risk of DNA damage, which is detrimental to retinal development and maintenance [8]. In addition to DNA damage sensitivity, metabolic demands in the retina are high, and exposure to oxidative stress is frequent [9]. As a result, strict transcriptional control is necessary for retinal cell efficiency. Research demonstrates that retinal cells develop abnormalities, like postmitotic cell degeneration, resulting from Top2b deletion [10]. Thus, the links between Top2b’s function and retinal defense against degeneration are furthered and the enzyme’s potential as a target for retinal health therapeutics is highlighted.

To understand the idiosyncrasies of retinal degeneration processes, mapping out the genes involved and tracking how they are regulated is necessary. We can uncover how transcription factors steer the disease’s course by closely organizing these genes and their activity. This catalog is important for determining where different retinal cell types are susceptible to degeneration. This review aims to identify the Top2b-linked genes expressed during retinal homeostatic processes, focusing on those vital for photoreceptor health. By exploring this relationship between retinal homeostatic dynamic gene expression and Top2b regulation, this study hopes to pull the curtain behind the regulatory mechanisms powering retinal maintenance and degeneration, identifying possible future research directions.

## 2. Approach

RNA sequencing data was obtained from Li et al.’s 2017 results [11], comprising retinal samples from wild-type and Top2b knockout (KO) mice. To detail the methodology of the study, at postnatal day zero (p0) and postnatal day six (p6), differential gene expression analysis (GSE86187) was conducted to identify genes with significant dynamic expression between the knockout and control. We acknowledge that this model represents a non-physiological extreme, as human BILU (B-cell immunodeficiency, limb anomalies, and urogenital malformations) syndrome patients retain partial TOP2B function [12]. Statistically significant gene expression differences (adjusted *p*-value < 0.05) were classified as “dynamically expressed” and included in the Top2b-linked genes list [11]. This list represents candidates regulated by Top2b during retinal development that have a possible role in retinal homeostasis. The 44 genes identified from this analysis should be considered as markers of transcriptional vulnerability rather than direct disease targets.

To further investigate the possible roles of Top2b-linked genes in retinal health regulation, dynamically expressed genes from Top2b KO mice were cross-referenced with existing retinal degeneration datasets. These datasets comprised RetNet, the Gene Ontology (GO) Browser, and a list of retinal degeneration-associated genes from Collin et al.’s 2020 [7] mice model study. These datasets were selected for their reliability and widespread acceptance in the retinal degeneration field. False Discovery Rate (FDR) calculations were performed, when possible, to validate cross-referencing. While FDR calculations were not applicable for RetNet, they are considered a trusted and comprehensive resource for retinal degeneration gene information. The Collin et al. mice model dataset was included as it provided statistically significant dynamic expression gene lists between genes with established relevance to retinal degeneration phenotypes.

Python 3.12 and the Pandas library were utilized to perform the gene list cross-reference. This was achieved by writing custom code to execute the matching of the Top2b-linked gene list from Li et al. 2017 [11] with the gene sets from RetNet, the Gene Ontology Browser, and Collin et al.’s mice dataset. All genes first had their gene symbols standardized, and the results were matched to determine which Top2b-linked genes were also associated with retinal degeneration in the literature. Dynamically expressed genes in the Top2b KO mice [11], present in at least one of the retinal degeneration datasets, were included in this study for pathway analysis. These genes were categorized based on their functional roles, such as visual transduction, ciliary function, signaling, and other impacted processes. Function labeling paralleled the gene annotations from public databases and retinal degenerative gene lists [7,13,14]. There was one sole categorization edit, merging the “Transcription Factors” and “Transcriptional Regulation” denominations. The pathway diagrams found in the Results section were all created by hand using Microsoft Visio and were based on a culmination of the literature for that particular gene.

To rectify the cross-referencing and data associations linking the Top2b differentially expressed genes to the genes in the retinal datasets, False Discovery Rates (FDRs) were calculated. FDR calculations were possible for the Gene Ontology Browser and Li et al.’s 2017 RNA-Seq gene sets [11], where all 44 genes had FDR values less than 0.05, strengthening the statistical significance in mapping these regulatory connections.

Top2b-linked retinal homeostatic gene pathway information was obtained from Reactome and an in-depth literature review. This was partly achieved using the Reactome API and UniProt IDs to retrieve detailed pathway data when available. The resulting 44 Top2b-linked genes, dynamically expressed in retinal degeneration, were analyzed using custom Python code to query Reactome. This analysis yielded pathway classification and the flow of each pathway. The pathways identified through Reactome were divided into the functional categorization of genes and used to establish connections between Top2b regulation, retinal degeneration, and specific pathways. After consulting the literature and/or Reactome, pathways were illustrated to visualize the link between Top2b and the possible signal cascade in the retina from regulatory errors.

## 3. Results

A total of 44 Top2b-linked dynamically expressed retinal homeostatic genes were identified and categorized based on retinal function (Appendix A Table A1), with “Ciliary Function and Trafficking” being the largest category with six genes (Figure 1A). This prevalence emphasizes the importance of ciliary dynamics with regard to photoreceptor and retinal health. Other major categories included, but were not limited to, “Signaling” and “Metabolism,” with five genes.

Gene expression analysis highlighted clear patterns across prenatal day data. Of the 44 Top2b-linked RD-associated genes, 30 were specifically expressed at postnatal day 0 (p0) and 13 at postnatal day 6 (p6) (Figure 1B). One gene, Glo1, was shared between the two time points (Appendix A Table A2). P0 is a time of rapid cell proliferation and early differentiation, while p6 is a stage where many cellular structures and functions, like photoreceptor development, are actively maturing. This p0 predominance suggests that transcription occurs early in postnatal retinal development, with fewer genes expressed in the later stages. Focusing on specific genes within these categories and their regulation is necessary to further the biochemical understanding of underlying retinal homeostasis and its pathways.

To assess the clinical relevance of our list of Top2b-linked genes, we cross-referenced our 44 genes with human retinal disease databases (Appendix A Table A3). Based on RetNet and the published literature, 21/44 (48%) genes have documented IRD associations, 4/44 (9%) are associated primarily with AMD, 2/44 (4.55%) are linked to both, and 17/44 (39%) do not fall into a straightforward IRD or AMD association. This shows that despite the extreme nature of the knockout model, it successfully identifies genes relevant to human retinal pathology.

The functional categorization reveals critical interconnected pathways through which Top2b regulates retinal homeostasis. First, ciliary function and trafficking (six genes including Bbs7), as the largest category, highlights Top2b’s essential role in maintaining the specialized connecting cilium structure of photoreceptors. This is crucial for protein transport between inner and outer segments, where disruption leads to the protein mislocalization, photoreceptor dysregulation, and visual impairment seen in ciliopathies like Bardet–Biedl syndrome. Additionally, DNA repair, RNA biogenesis, and protein modification (four genes including Ubb) demonstrate Top2b’s protective function against oxidative damage, a constant threat in the metabolically active retina. These genes are critical for maintaining genomic stability, proper protein folding, and the subsequent clearance of damaged protein through the ubiquitin–proteasome system. Thus, disruption in this pathway results in protein aggregation and cellular stress responses. Furthermore, visual cycle and retinoids (two genes including Rbp4) directly connect Top2b to the light-sensing biochemical pathways by facilitating vitamin A transport and metabolism into 11-cis-retinal, the chromophore vital for phototransduction. Dysfunction in this role results in disruption of the visual cycle, which leads to impaired photoreceptor function and degeneration. Moreover, cytoskelton dynamics (two genes including Cetn2) reveal the influence of Top2b on structural integrity and cellular organization within photoreceptors by maintaining basal body anchoring, axoneme stability, and proper microtubule organization in the connecting cilium. Mutations in this functional category cause the disorganization of the outer segments and compromised structural integrity. Similarly, the signaling (five genes including Pik3r1) distinction links Top2b to critical neuroprotective pathways that defend against a myriad of stressors via the activation of phosphoinositide-mediated signaling cascades. These cascades notably promote photoreceptor survival through the inhibition of apoptotic pathways, with dysfunction increasing susceptibility to stress-induced degeneration. Lastly, transcription factors and regulation (four genes including Crx) illustrate Top2b’s regulatory role over broad developmental and maintenance gene networks. This is achieved by controlling the expression of hundreds of photoreceptor-specific genes, coordinating rod–cone differentiation, and regulating phototransduction. These mutations cause developmental defects and progressive retinal degeneration.

### 3.1. Ciliary Function and Trafficking

The apparent transcriptional regulation of Bbs7 by Top2b spotlights both genes’ critical role in maintaining ciliary function. Top2b facilitates the expression of long genes, like that of Bbs7, which encodes proteins vital for a process integral to photoreceptor health: ciliary trafficking. Our published RNA-Seq results of the Top2b KO mice photoreceptors showed the significant downregulation of Bbs7 in the KO mice compared to the wild type [11]. Top2b disruption may hinder Bbs7 transcription, leading to protein mislocalization within ciliary structures. This dysfunction can exacerbate cellular stress and accelerate retinal degeneration, exposing a potential molecular mechanism underlying Bbs7-associated diseases (Figure 2).

Bbs7 is a component of the Bardet–Biedl syndrome protein complex (BBSome), a pivotal facilitator of ciliary function regulation and protein trafficking within photoreceptors [15]. The BBSome controls the transport of membrane proteins to and from the primary cilium, with the caveat that disruptions in this pathway often lead to retinal degeneration through impaired protein localization and ciliary dysfunction [16].

BBSomes in retinal photoreceptors also exhibit unique properties compared to BBSomes elsewhere in the body [17]. The rules of entry for the cilia of photoreceptors can differ from that of other primary cilia, with the photoreceptor cilia allowing the admittance of partially assembled BBSomes and the primary cilia allowing only fully assembled BBSomes to enter [17].

**Figure 2 cells-14-00887-f002:**
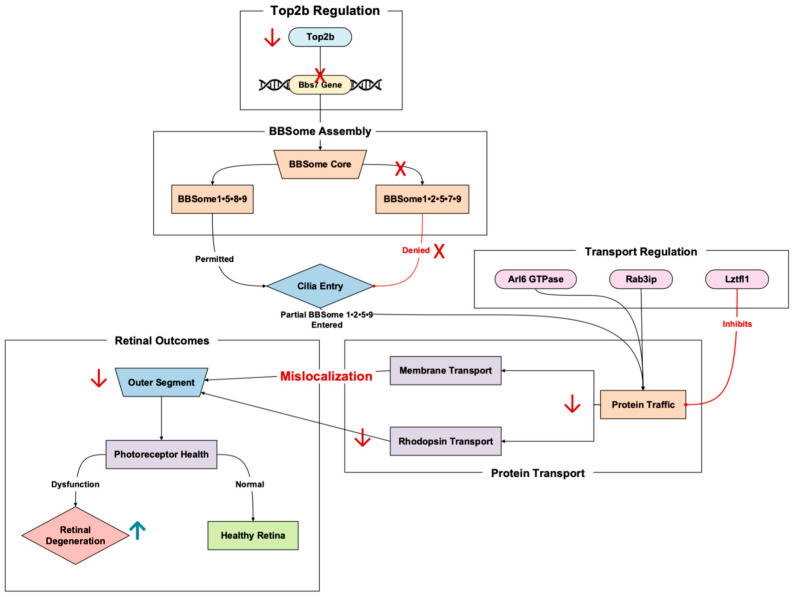
The Top2b-regulated gene, Bbs7, is an essential protein for BBSome-mediated ciliary trafficking. Pathway based on Chandra et al., 2022 [17], Hsu et al., 2021 [18], Milacic et al., 2024 [14], Li et al., 2017 [11], and Weinbrecht et al., 2017 [16]. Bbs7 mutations cause shortened outer segments and cone–rod dystrophy. Red arrows denote downregulation, while red crosses in this figure denote the absence of promoter binding and inhibition. The upward blue arrow represents upregulation.

The Top2b-linked Bbs7 influences whether a BBSome complex can enter cilia. The two currently known BBSome subcomplexes, comprising the Bbs gene with the corresponding numbers identified, are BBSome1•5•8•9 and BBSome1•2•5•7•9, with the possibility of the former allowing ciliary entry and the latter denying ciliary entry [17]. In Bbs7 knockout mice, BBSome1•2•5•7•9 does not materialize, but an incomplete version sans the 7, BBSome1•2•5•9, does. This incorrect, partial supercomplex can gain entry while its proper form in the wild type is, again, denied [18]. This partial BBSome entrance results in mislocalization occurring in the photoreceptors of the knockout mice, demonstrating the important role Bbs7 has in proper trafficking recognition and movement [18]. As a result, the Bbs7 knockout mice exhibited retinal degenerative symptoms.

The pathway involving Bbs7 includes several critical steps that highlight its role in trafficking proteins to the primary cilium. Arl6, a GTPase, interacts with the BBSome to target specific cargo proteins to the primary cilium [18]. The complex binds to Rab3ip and associates with ciliary cargo, ensuring proper localization [14]. However, disruptions occur when Lztfl1 binds to the BBSome, thereby preventing its trafficking to the cilium and disrupting protein delivery [16]. This signal cascade drives the proper functioning of photoreceptor cells, with any failure in these steps possibly resulting in protein mislocalization and the degeneration of photoreceptor outer segments [15].

Bbs7 dysfunction involves extensive structural abnormalities, such as shortened outer segments and reduced rhodopsin transport (key contributors to photoreceptor cell death) [16]. Patients with Bbs7 mutations experience cone–rod dystrophy, manifesting in early cone dysfunction followed by rods as well [15]. This outcome is corroborated by animal models and clinical study findings, where central cone dysfunction has led to visual field constriction and subsequent rod photoreceptor loss. Structural imaging revealed the thinning of the outer nuclear layer and disruptions in key areas, both signatures of photoreceptor degeneration linked to ciliary dysfunction [15].

The regulation of Bbs7 by Top2b may provide an insight into its transcriptional dynamics. As a regulator of long genes, Top2b is crucial for enabling the expression of genes like Bbs7 that encode proteins essential for complex, tissue-specific processes such as ciliary trafficking [11]. Disruption of Top2b activity could impair the expression of Bbs7, exacerbating protein mislocalization and catalyzing retinal degeneration.

### 3.2. DNA Repair, RNA Biogenesis, and Protein Modification

The polyubiquitin gene Ubb is promoted by Top2b upregulation, a critical player in retinal cellular maintenance through protein degradation and DNA repair [11]. Ubiquitin, encoded by Ubb, tags damaged or misfolded proteins for degradation through the ubiquitin–proteasome system (UPS) [19]. Notably, Ubb expression is significantly higher than the other polyubiquitin in the retina, Ubc, meaning Ubb is the dominant ubiquitin facilitator [20]. In photoreceptors, this process is especially crucial considering the importance of the cell in oxidative stress exposure and ensuring the proper function of proteins involved in the visual cycle (Figure 3).

As demonstrated in Ubb knockout mouse models, Ubb disruption significantly reduces ubiquitin concentration [21]. Retinal degeneration may soon follow once ubiquitin concentration is decreased past a certain threshold. This manifested in the Ubb KO mice exhibiting progressive outer nuclear layer thinning, driven by considerable photoreceptor loss. This degeneration was also paired with decreased rhodopsin expression and increased glial fibrillary acidic protein (GFAP) expression, a signal of photoreceptor stress [21].

A key pathway linking Ubb to retinal homeostasis is the gene’s regulation of histone ubiquitination. Decreased ubiquitinated histone H2A (H2A-Ub) levels in Ubb KO impair transcriptional regulation, thus weakening the responses to DNA damage and oxidative stress [20]. This transcriptional dysregulation is especially detrimental to photoreceptors, which are highly metabolically active and exposed to constant oxidative stress from outer segment light exposure [9]. In addition to histone ubiquitination, the UPS plays a vital role in clearing damaged proteins and aggregates under stress. In Ubb-deficient retinas, failure to clear these protein aggregates exacerbates oxidative damage, leading to reactive glial cell activation [20]. The initial glial response is protective but becomes damaging when sustained. This glial “Jekyll and Hyde” dynamic further contributes to retinal tissue dysfunction and photoreceptor apoptosis. Notably, to compensate for the Ubb deficiency, the cell upregulates Ubc, another polyubiquitin gene [20]. However, this Ubc’s attempt at ubiquitin homeostasis is insufficient, and the retina continues to degrade.

It cannot be forgotten that the significance of Ubb in protecting photoreceptors from oxidative and transcriptional stress was found to be linked to Top2b regulation. The dysregulation of this pathway reveals a critical mechanism in retinal regulation, the ubiquitin–proteasome system, and DNA repair mechanisms.

### 3.3. Visual Cycle and Retinoids

Retinol-binding protein 4 (Rbp4) plays a crucial role in the visual cycle in its transport of retinol, also referred to as vitamin A, through the bloodstream into the retinal pigment epithelium (RPE) [22]. In the RPE, retinol is metabolized, producing 11-cis-retinal, a light-sensitive chromophore required for phototransduction in the rods and cones of photoreceptor cells [23]. The possible inhibitory regulation of Rbp4 by Top2b in photoreceptors ensures adequate retinoid delivery balance and homeostasis in the retina [11]. Dysregulation of this pathway has profound implications for retinal health (Figure 4).

Impaired retinoid transport has been shown to deplete chromophore levels and, as a result, power retinal degeneration via phototransduction damage [24]. Rbp4 knockout in an early study showed decreased 11-cis-retinal, which further led to lower rhodopsin levels and impaired phototransduction [11]. This was also exhibited in Montenegro et al.’s Rbp4 knockout mice, where significantly decreased 11-cis-retinal and other compounds were measured at four months. As the four-month age progressed into eight months, photoreceptor degeneration was apparent, demonstrating to researchers a possible link between retinoid deficiency and retinal cell death [24].

In addition to impaired transport, low retinoid concentration was shown to inherently activate inflammatory and oxidative stress pathways in RPE cells, resulting in photoreceptor death [24]. The researchers examined the concentration of lipofuscin, a visual cycle byproduct, in RPE cells. Notably, lipofuscin contains A2E, a toxic bis-retinoid that can assist in the triggering of inflammation and oxidative stress [25]. However, interestingly enough, the Rbp4 knockout mice had a more than 70% reduction in A2E levels and yet a significant inflammatory and oxidative stress response to the point of retinal degeneration [24]. This suggests that the previous notion that A2E may be a big driver in triggering inflammation and oxidative stress is too simplistic. Montenegro et al. showed that the retinoid level itself is a critical factor in influencing oxidative damage, inflammation, and, subsequently, retinal degeneration, and not just A2E.

In terms of retinal structure, transmission electron microscopy imaging and histology displayed progressive outer nuclear layer thinning in the Rbp4 knockout mice [11,24]. Since the photoreceptors are the retinal cell type that dominates the outer nuclear lining, a thinner outer nuclear layer is a damning representative of photoreceptor apoptosis. Researchers also witnessed fewer synapses in a crucial retinal layer where synaptic connections occur between visual signal processing cells (the inner plexiform layer), a sign that signal transmission from photoreceptors to other retinal cells could be impaired due to Rbp4 absence [11].

### 3.4. Cytoskeleton Dynamics

Cetn2, a calcium-binding protein, plays a major role in maintaining the structural stability and function of the connecting cilium in photoreceptor cells [26]. The Top2b-regulated gene encodes a calcium-binding protein that localizes to the centrosome, basal body, and cilium transition zone, critical areas for ciliary trafficking and axoneme stability [27]. Li et al.’s 2017 RNA-Seq results of Top2b KO mouse photoreceptors showed the significant downregulation of Cetn2 in the KO mice compared to the wild type [11]. It is worth noting that disruptions in Cetn2 expression solely do not have extensive implications for retinal degeneration due to Cetn3 redundancy [28]. However, the dysregulation nonetheless results in retinal degenerative phenotypes, especially that of photoreceptor axonemes and connecting cilium destabilization [26].

Double Cetn2 and Cetn3 knockouts lead to early onset degeneration, manifesting in a decreased connecting cilium length, axoneme disorganization, and outer segment malformations [27]. These phenotypes culminate in photoreceptor apoptosis and outer nuclear layer thinning, with almost complete degeneration when left to worsen [26]. Cetn2-involved pathways include the gene’s role in centrosome-associated processes such as basal body anchoring to the plasma membrane and the recruitment of mitotic centrosome proteins and complexes [29]. Specifically, Cetn2′s interaction with other centrosome proteins ensures proper cilium assembly and attachment to the plasma membrane (Figure 5). When Cetn2 is disrupted, these crucial cellular processes fail, resulting in damaged photoreceptor function. In addition to anchoring and recruiting, Cetn2 regulates PLK1 activity during G2/M transition (a pathway essential for cell cycle progression and centrosome stability) [27]. Therefore, Cetn2 loss can alter regulation and spurring defects in ciliary maintenance and photoreceptor survival (Figure 5).

Another important Cetn2 mechanism linked to degeneration is Spata7 depletion. Spermatogenesis-associated 7 (Spata7) is a pivotal photoreceptor-specific distal connecting cilium organizer whose function is important in retinal health. Loss of Spata7 results in microtubule destabilization, radial expansion of the axoneme, and the mislocalization of proteins, all things which are necessary for phototransduction [30].

Overall, Cetn2 regulation of Top2b illustrates the significance of photoreceptor health [11]. Disruptions in the Cetn2 pathway, whether through genetic mutations or impaired transcriptional control, highlight a critical point of retinal homeostatic regulation. These studies suggest that targeting ciliary trafficking and axoneme stabilization may offer therapeutic potential for photoreceptor dysfunction.

### 3.5. Signaling

In rod photoreceptors, Pik3r1 encodes the p85α regulatory subunit of phosphoinositide 3-kinase (Pi3k) [31]. The Top2b-regulated gene has a central role in retinal neuroprotective signaling pathways. Pi3k, once activated by receptor tyrosine kinases (like insulin receptors), helps generate phosphoinositides like Pip3 (Figure 6). These phosphoinositides go on to activate downstream Akt signaling cascades. These cascades promote photoreceptor cell survival by inhibiting apoptotic pathways and maintaining retinal cellular homeostasis [32]. Pik3r1 promotion by Top2b optimizes transcriptional control of this critical signaling gene in photoreceptors. In Pik3r1 knockout mouse models, altering Pik3r1 leads to an inability to recover phototransduction and increased susceptibility to retinal degeneration induced by stress [7].

In the context of stress triggered by light, the Pi3k-Akt pathway becomes particularly important due to the protection photoreceptors receive from the pathway’s activation. In Pik3r1 knockout models, the absence of the p85α subunit results in the slower trafficking of arrestin, a protein necessary for phototransduction termination [33]. Typically, arrestin is trafficked from the inner segments to the outer during light adaptation, thus delaying phototransduction and increasing photoreceptor stress. In addition to slow arrestin translocation, impaired Pi3k signaling disrupts cytoskeletal stability [33]. This arises because rod outer segment disorganization further catalyzes degeneration (Figure 6).

Rajala et al.’s 2022 study continues to validate this reality [31]. The researchers demonstrate that Pi3k-Akt signaling is imperative for photoreceptor survival in the retina, elaborating that the loss of Pik3r1 disrupts PI3K signaling, which reduces Akt phosphorylation and leads to increased oxidative stress and apoptosis in photoreceptors [31]. The study also looks more into the role of phosphoinositide phosphatases in Pi3k signaling and retinal degeneration. Several Pi3 phosphatases can antagonize the Pi3k pathways, whether Pten or Mtmr4. When these phosphatases are upregulated in rods, researchers point to an upregulation in the suppression of PI3K signaling [31]. This suppression may spur retinal degenerative phenotypes because of the possible exacerbation of Pik3r1 loss effects.

Through Top2b promoting the expression of long genes known for their signaling, like that of Pik3r1, neuroprotective pathways activate when retinal cells react to stress [11]. Top2b activity alterations impair this transcriptional regulation, reducing the retina’s ability to respond to environmental and oxidative stressors, thus promoting retinal degeneration. By maintaining Pik3r1 transcription, Top2b supports Pi3k/Akt signaling, reinforcing its role in preventing photoreceptor apoptosis and maintaining retinal function.

### 3.6. Transcription Factors and Regulation

Crx (cone–rod homeobox) is an influential transcription factor essential for the development and differentiation of photoreceptors. It regulates the expression of a variety of photoreceptor-specific genes, from phototransduction genes to retinal metabolic genes [34]. In transcription, the role of Crx is tightly regulated by Top2b, meaning tampering with this pathway can lead to transcriptional failure and subsequent retinal degeneration down the line.

Crx holds a multifaceted role in photoreceptor development, with influences a myriad of processes. Crx does not act alone, with the transcription factor interacting with Otx2 to regulate important photoreceptor genes. This is necessary to note, because this Crx-Otx2 interaction can be a source of early-onset retinal degeneration [35]. Upstream mutations of proteins that interact with Crx-Otx2 lead to gene activation and repression imbalances. This imbalance can result in the activation of photoreceptor genes at the wrong time and the downregulation of crucial photoreceptor genes, resulting in photoreceptor dysfunction and apoptosis [35].

Crx functions with other transcription factors, like Nrl and Nr2e3, to promote rod gene expression while suppressing cone gene expression during differentiation [34]. Loss of Crx disrupts this balance, damaging rod development and maintenance while activating cone-specific genes (Figure 7). Crx mutations lead to disorders such as Leber congenital amaurosis (LCA) and cone–rod dystrophy, both caused by progressive photoreceptor loss and retinal degeneration [36]. In one study, Crx knockout models exhibited reduced key rod-specific gene expression, like that of Rho (rhodopsin), and the disorganization of the outer nuclear layer structure [37]. In another, more recent Crx knockout model study, researchers replaced the Crx allele with a fluorescent marker. Using human embryonic stem cells, these cells were differentiated into retinal organoids to model human retinal development [36]. The Crx haploinsufficiency led to the human retinoid having outer nuclear layer formation delayed by almost double the time it took the normal retinal organoid to form. In addition to the ONL delay, researchers also saw a loss of photoreceptor outer segments, resulting in the impairment of the phototransduction process [36]. To elaborate further on the outer nuclear layer, live-cell imaging showed that the Crx monoallelic knockout had reduced movement efficiency and irregular migration patterns [36]. Notably, there was a delay in moving to their correct position in the ONL, with increased numbers of mislocalized cells. All these impaired processes due to Crx dysregulation can culminate in an individual experiencing early-onset retinal degeneration (Figure 7).

Pathways linking Crx to retinal degeneration have also been studied, allowing us to paint a bigger picture of Top2b’s role in retinal homeostasis and its cascade effects. Peng et al. outline the example of Crx binding to the promoter of Pde6b, a gene essential for cGMP hydrolysis in phototransduction and a gene that is also one of the 44 Top2b-linked retinal homeostatic genes outlined in this review (Table A1). In rod photoreceptors, Crx works with Nrl to enhance Pde6b expression [37]. This ensures that the cell is experiencing proper phototransduction and cellular function. In Crx-deficient conditions, Pde6b becomes downregulated and increases cGMP levels, thus triggering calcium excess and apoptosis in photoreceptors (Figure 7). In addition to Nrl, Crx can work with Nr2e3 to repress cone-specific genes, allowing the rod phenotype to dominate in the retina [37]. If this cone repression did not occur, photoreceptor cell development regarding the cell type identity would be disrupted, triggering apoptosis and retinal degeneration [37]. Pan et al.’s 2023 study echoes and builds upon these results with transcriptomic analysis and live-cell imaging, among other methodologies [37]. The researchers also saw altered rod–cone balance in their Crx monoallelic knockout retinal organoids. They noted that rod photoreceptor gene expression recovered while the cones did not. This led the researchers to conclude that Crx plays a more decisive role in cone differentiation, potentially explaining the cone-distinguished degeneration in Crx-linked retinal diseases. In addition to photoreceptor cell type balance, Crx was shown to interact with actomyosin networks in the retinal organoid studies, which further reinforces the transcription factor’s role in cell movement and ONL formation [36]. It is also worth noting how overactive cytoskeletal contraction blocked proper photoreceptor placement, disrupting the specialized layered structure that the retina requires.

As a long-gene regulator, Top2b is paramount in promoting the Crx transcription regulation of its downstream targets [11]. Top2b dysfunction impairs the transcription of Crx-dependent genes, thus compromising photoreceptor function, emphasizing the importance of its interactions in sustaining retinal health and preventing retinal degenerative scenarios. The outlined Crx pathway illustrates how transcriptional dysregulation can lead to retinal degeneration while demonstrating the possible target of Top2b-Crx for treating transcriptional defects in retinal diseases.

## 4. Discussion

Intricate regulatory networks hold together the complex structure that makes up the retina to ensure its function and survival. Looking at the proper localization of proteins to and from the photoreceptor outer segments, genes like Bbs7 and Cetn2 reveal how disruptions in their regulation can lead to cellular disorganization and degeneration. Other genes like Rbp4 are critical genes that hold roles that emphasize the defense of photoreceptors against oxidative stress and neuroinflammation while assisting in phototransduction. In addition to the visual cycle, signaling and DNA repair genes, like Pik3r1 and Ubb, spotlight pathways that respond to cellular stress and damage by activating neuroprotective signaling cascades or clearing damaged proteins. Lastly, broad transcription factor genes like Crx illustrate the profound impact of transcriptional regulation on essential processes, ranging from phototransduction via interactions with Pde6b to regulating rod- and cone-specific gene expression vital for photoreceptor differentiation and survival. All of these genes, powered by the regulatory activity of Top2b [11], can dictate the health of an individual’s retina, with all of these genes also holding the power to trigger retinal degeneration when its expression goes awry.

While our findings derive from complete Top2b knockout models, it is important to note that human TOP2B mutations present differently. Patients with BILU syndrome carry partial loss-of-function mutations yet exhibit no documented ophthalmologic issues [12]. This discrepancy suggests that complete versus partial Top2b loss may have distinct effects on retinal tissue, and that compensatory mechanisms may exist in humans with partial function. Future studies utilizing conditional knockdown or hypomorphic alleles that better model human mutations would provide more clinically relevant insights.

The broader implications of these findings center on the potential of targeting Top2b and its regulatory pathways for areas of therapeutic attention. By identifying genes and pathways dependent on Top2b activity, this study provides a foundation for developing strategies to understand transcriptional dysregulation and to work to preserve retinal function. This could lead to developing retina-specific delivery systems, biomarker identification for early intervention, and combination therapies targeting multiple pathways simultaneously.

The clinical translation potential of our findings are considerable. The 44 Top2b-regulated genes identified represent promising diagnostic biomarkers for the early detection of retinal degeneration before irreversible photoreceptor loss occurs. Specifically, proteins encoded by Bbs7, Ubb, and Rbp4 could serve as non-invasive indicators of retinal health. Genetic screening panels targeting these genes might identify at-risk patients before symptom onset, enabling preventive interventions. The disease segregation shown in Appendix A Table A3 also holds important therapeutic implications. IRD-associated genes represent candidates for gene replacement therapy, while AMD-associated genes suggest targets for anti-inflammatory or metabolic support strategies. The shared genes (Pik3r1, Rbp4) may explain the overlapping pathophysiology in some patients. In addition, the diverse functional categories we have identified suggest that combination therapies addressing multiple pathways (ciliary trafficking, protein degradation, and neuroprotective signaling) may prove more effective than single-target approaches.

While numerous pathways and genes have been implicated in retinal degeneration, the explicit pathway flows between Top2b were solely based on published data from GSE86187, RetNet, and the Gene Ontology Browser. Though valuable resources, all pathway interactions were not experimentally validated outside the literature confirmation and were only statistically validated by False Discovery Rate calculations for the GSE86187 and Gene Ontology Browser datasets. With limitations in establishing causation versus correlation in the observed interactions, widely accepted retinal degeneration databases for cross-referencing were utilized and backed up with FDR calculations when possible. Several additional methodological limitations warrant attention. Our bioinformatic approach may have had inherent biases toward well-characterized genes, potentially underrepresenting novel regulatory elements in the Top2b network. The analysis was also constrained to specific development timepoints (p0 and p6), limiting our understanding of Top2b’s influence throughout complete retinal development and adult homeostatic maintenance. Additionally, functional categorization required subjective decisions for genes with overlapping effects, which may influence interpretations of pathway importance. The absence of retinal phenotypes in BILU syndrome patients and the embryonic lethality of complete TOP2B loss raise important questions about our findings’ translational relevance. Rather than suggesting TOP2B mutations cause retinal disease, our analysis reveals which retinal genes depend on Top2b for normal expression. These dependencies may become relevant when other factors, such as aging, oxidative stress, and mutations in the regulated genes themselves, compromise retinal health. The value lies not in modeling TOP2B diseases, but in identifying vulnerable points in retinal transcriptional networks that could be therapeutically supported when stressed by other mechanisms.

Looking towards the future, several promising research directions could address these limitations and advance clinical applications. Single-cell transcriptomics could reveal cell-type-specific Top2b regulation patterns, while epigenetic profiling through ChIP-seq and ATAC-seq would clarify direct binding sites and chromatin accessibility changes. In regard to the direct causality limitation, CRISPR-based validation of key regulatory relationships could establish causality in the Top2b network, while in another sense, pharmacological studies targeting Top2b modulation could assess potential therapeutic benefits in retinal degeneration models.

The visual cycle pathway involving Rbp4 already demonstrates therapeutic promise, with high-dose vitamin A supplementation showing benefits in related retinal dystrophies [22]. For ciliopathies involving Bbs7, compounds that enhance BBSome assembly could possibly prevent the protein mislocalization that drives photoreceptor death. The ubiquitin–proteasome pathway regulated by Ubb offers targets for the protease-enhancing compounds currently in trials for other neurodegenerative conditions [38]. By leveraging these existing therapeutic avenues and fostering collaboration between clinical researchers and pharmaceutical developers, the path from these discoveries to clinical implementation could be significantly accelerated.

## 5. Conclusions

This literature review explores topoisomerase II beta (Top2b)’s critical role as a key regulator of 44 uncovered genes involved in retinal health processes. Processes like ciliary function, metabolism, DNA repair, and transcriptional regulation allow new insights to illuminate its involvement in retinal homeostasis. By regulating the expression of influential genes across various functional categories, the genes analyzed in this review, Bbs7, Ubb, Rbp4, Cetn2, Pik3r1, and Crx, highlight the broad scope of Top2b’s regulatory activity. Future studies should focus on Top2b and its functionality under stress in retinal cells using advanced tools like single-cell RNA sequencing and possible CRISPR-based editing. Longitudinal studies in both animal models and human patients are also needed to uncover the intricate details of Top2b’s involvement in retinal degeneration. Future research can demystify the complex interactions between Top2b, retinal health, and gene cascading by combining experimental and computational approaches.

Given Top2b’s essential role in retinal homeostasis, therapeutic strategies should focus on modulating specific downstream pathways rather than targeting Top2b directly. For instance, enhancing compensatory pathways (such as upregulating functional BBSome assembly in Bbs7-deficient states) or providing pathway-specific supplements (like antioxidants for Nxnl2 pathway support) represent more viable therapeutic approaches. Additionally, gene therapy targeting specific Top2b-regulated genes showing reduced expression could possibly restore function without affecting global Top2b activity. This allows for new avenues for the development of targeted treatments. Several of these approaches have a precedent in clinical development: vitamin A supplementation for RBP4-related retinopathies is already in use [22], proteasome modulators like bortezomib are FDA-approved for other indications and could be repurposed, and BBSome-stabilizing compounds are in preclinical development for ciliopathies [16]. Such therapeutic interventions could potentially slow or halt the progression of diseases such as RP and AMD, significantly impacting patient care and quality of life. Ultimately, this review provides a more transparent understanding of how Top2b is connected to retinal regulation by finding, categorizing, and analyzing retinal homeostasis-associated genes known to be linked to Top2b. This all culminates in providing a draft blueprint for future research to gain further insight into Top2b’s impact on retinal degenerative processes, giving way to new therapeutic potential.

## Figures and Tables

**Figure 1 cells-14-00887-f001:**
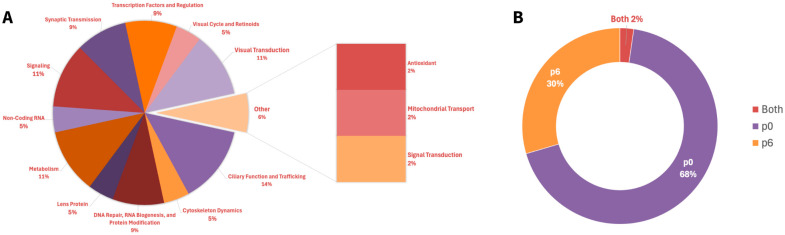
Breakdown of the 44 identified Top2b-linked genes also dynamically expressed in retinal homeostasis. (**A**) Genes are divided by function in the retina. (**B**) Top2b-linked dynamic expression by postnatal day, as reported from Li et al.’s 2017 Top2b KO mice retina model study [11].

**Figure 3 cells-14-00887-f003:**
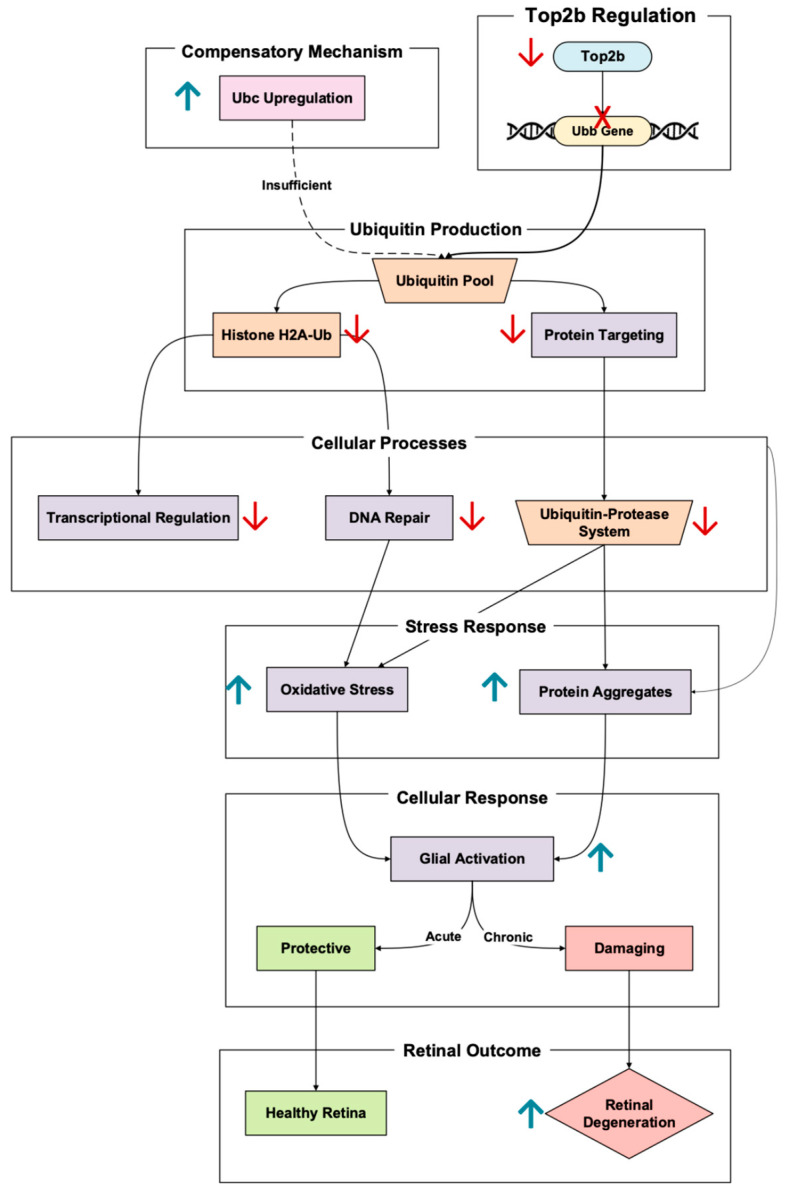
Top2b-regulated gene, Ubb, is a key player in protein degradation and transcriptional regulation in the retina. The pathway is based on Lim et al., 2019 [20], Li et al., 2017 [11], and Lobo et al., 2021 [9]. Ubb deficiency leads to protein aggregation, reactive gliosis, and photoreceptor death. Red arrows denote downregulation, while the red cross in this figure denotes the absence of promoter binding. The upward blue arrow represents upregulation.

**Figure 4 cells-14-00887-f004:**
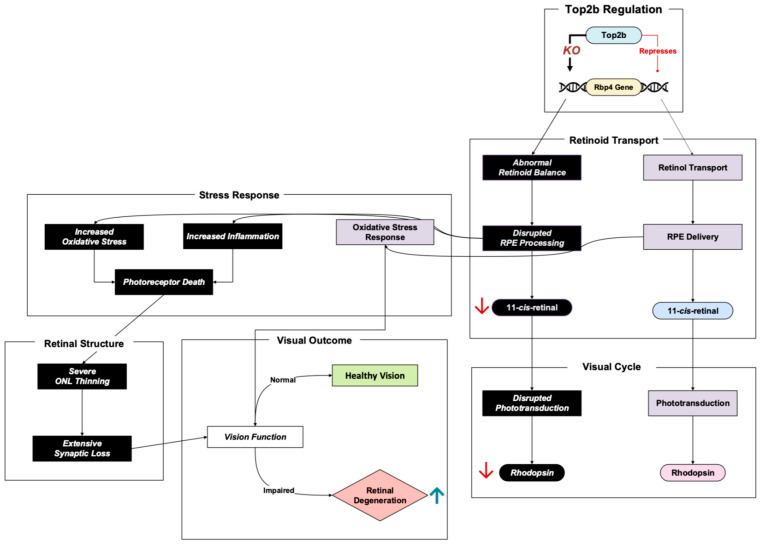
Top2b-regulated gene, Rbp4, is crucial for retinoid transport and visual cycle. Pathway based on Steinhoff et al., 2021 [23], Li et al., 2017 [11], and Montenegro et al., 2022 [24]. Rbp4 dysregulation and retinoid deficiency result in consequences for rhodopsin levels, oxidative stress response, and retinal structure. Boxes in black represent the Top2b-impaired pathway; red arrows denote downregulation; while upward blue arrows represent upregulation.

**Figure 5 cells-14-00887-f005:**
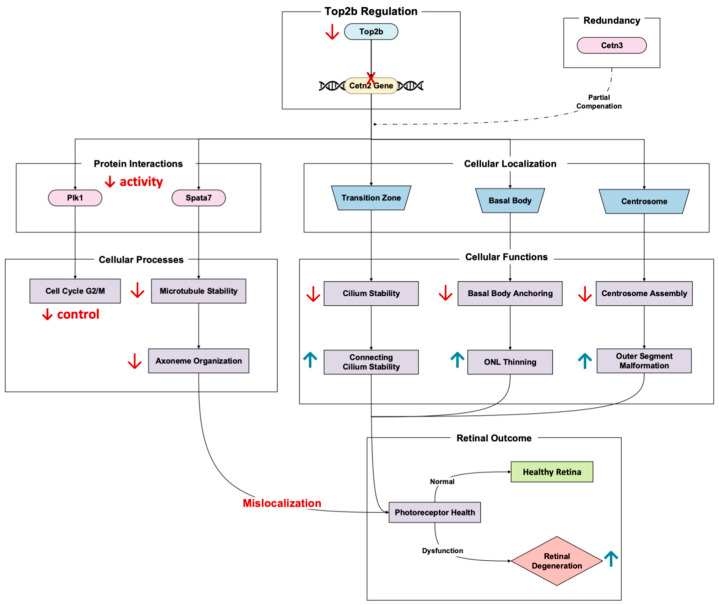
Top2b-regulated gene, Cetn2, maintains photoreceptor ciliary structure and function. Pathway based on Ying et al., 2019 [27], Baehr et al., 2019 [28], Ning et al., 2021 [26], Li et al., 2017 [11], and Lu et al., 2022 [30]. Cetn2 deficiency causes shortened connecting cilia, axoneme disorganization, and photoreceptor degeneration. Red arrows denote downregulation, while the red cross in this figure denotes the absence of promoter binding. The upward blue arrow represents upregulation.

**Figure 6 cells-14-00887-f006:**
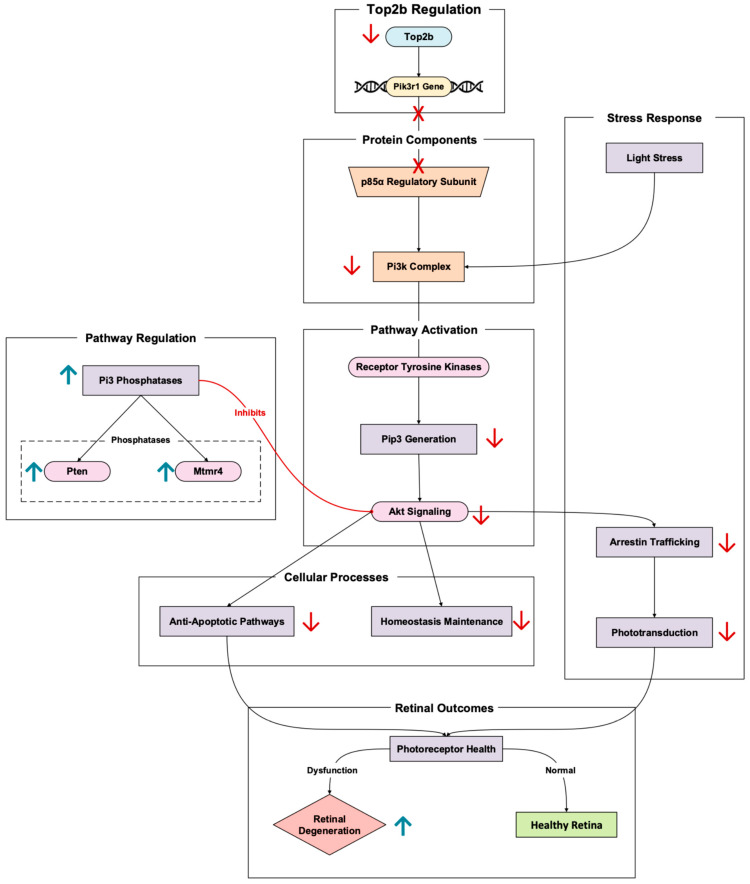
Top2b-regulated gene, Pik3r1, represents a pivotal signaling cascade in photoreceptor cells. Pathway based on Zheng et al., 2018 [32], Collin et al., 2020 [7], Ivanovic et al., 2011 [33], Rajala et al., 2022 [31], and Li et al., 2017 [11]. Pik3r1 dysregulation lowers the retina’s ability to handle environmental and oxidative stressors, increasing the risk of retinal degeneration. Red arrows denote downregulation, while red crosses in this figure denote the absence of protein. The upward blue arrow represents upregulation.

**Figure 7 cells-14-00887-f007:**
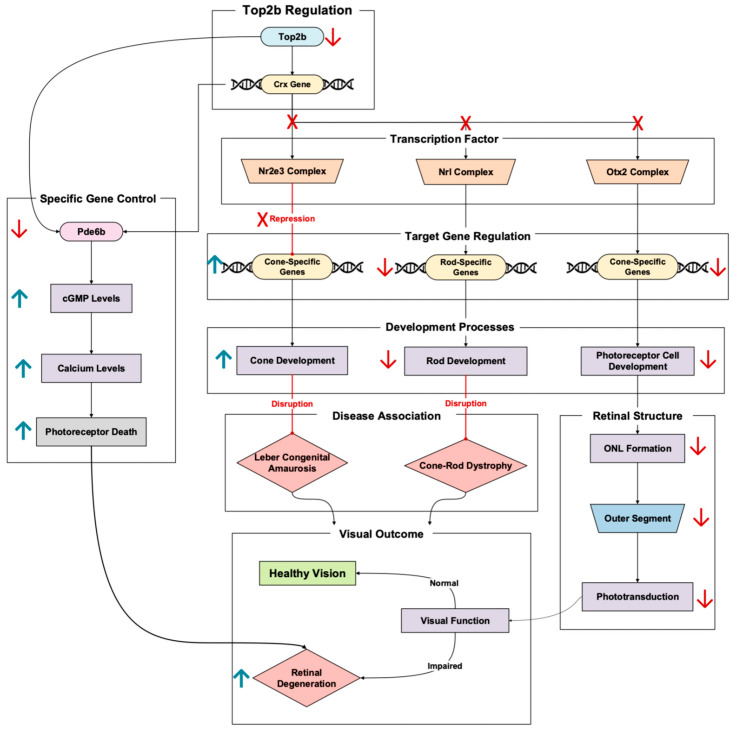
Top2b-regulated gene, Crx, is a master regulator of photoreceptor development and maintenance [8]. Pathway based on Langouet et al., 2022 [35], Zheng & Chen., 2024 [34], Pan et al., 2023 [36], and Peng et al., 2005 [37]. Top2b regulation of Crx represents a critical control point for photoreceptor development, differentiation, and survival. Red arrows denote downregulation, while red crosses in this figure denote a lack of protein binding and a lack of repression due to absence. The upward blue arrow represents upregulation.

## Data Availability

The data used to support this study are derived from publicly available databases and repositories. Genomic data analysis utilized the Gene Ontology Browser (http://geneontology.org/, accessed on 3 September 2024), RetNet Database (https://sph.uth.edu/retnet/, accessed on 3 September 2024), and Reactome Pathway Database (https://reactome.org/, accessed on 27 September 2024). Differential gene expression analysis was based on RNA sequencing data from Li et al. 2017 [11], which is publicly accessible through the Gene Expression Omnibus (GEO) database under accession number GSE86187. Custom Python code used for cross-referencing gene lists and performing pathway analysis values is available upon reasonable request from the corresponding author.

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
