# Peer review of "Top2b-Regulated Genes and Pathways Linked to Retinal Homeostasis and Degeneration"

_cells, 2025, doi:10.3390/cells14120887_

Round 1
Reviewer 1 Report
Comments and Suggestions for Authors
This is a well-organized and comprehensive review that summarizes current knowledge regarding Topoisomerase II beta (Top2b) and its regulatory role in retinal homeostasis and degeneration. The authors utilized RNA-Seq data and cross-references with RD datasets to identify 44 Top2b-associated genes implicated in key retinal functions, including ciliary trafficking, transcriptional regulation, DNA repair, and the visual cycle. Mapping these genes provides valuable mechanistic insights that enhance our understanding of Top2b’s role in retinal development and disease.
Major:
1. A significant limitation of prior studies on Top2b in retinal function is that they have primarily relied on mouse models with complete gene deletions (also cKO in retinal progenitor cells). This distinction is critical, as patients with partial loss-of-function mutations in TOP2B (known as BILU syndrome), exhibit B-cell developmental defects, distal limb anomalies, and urogenital malformations, yet no ophthalmologic abnormalities have been reported. The discrepancy between complete loss in animal models and partial loss in humans requires more discussion.
2. Figures 2 through 7 appear highly similar in format and content. These could be consolidated into one or two major figures that integrate multiple gene pathways or clusters.
3. Certain statements in the discussion are unclear and potentially contradictory. For example, lines 549–551 "targeting Top2b or its downstream pathways could be therapeutic for retinal degenerative diseases". However, if Top2b is essential for maintaining retinal homeostasis, this strategy may be counterintuitive. Greater clarification is needed regarding whether the therapeutic goal is to enhance, inhibit, or modulate Top2b function, and under what specific pathological conditions.
Minor:
Please review the manuscript for redundancy. For instance, lines 47–49 repeat similar points about the role of photoreceptors in initiating vision.
Author Response
Comments 1: A significant limitation of prior studies on Top2b in retinal function is that they have primarily relied on mouse models with complete gene deletions (also cKO in retinal progenitor cells). This distinction is critical, as patients with partial loss-of-function mutations in TOP2B (known as BILU syndrome), exhibit B-cell developmental defects, distal limb anomalies, and urogenital malformations, yet no ophthalmologic abnormalities have been reported. The discrepancy between complete loss in animal models and partial loss in humans requires more discussion.
Response 1: Thank you for this important observation. We agree that the discrepancy between complete Top2b loss in mouse models and partial loss-of-function mutations in humans requires clarification. We have revised the manuscript to acknowledge this limitation and discuss its implications.
In the Approach section (page 3)
"We acknowledge that this model represents a non-physiological extreme, as human BILU syndrome patients retain partial TOP2B function [11]. [...] The 44 genes identified from this analysis should be considered as markers of transcriptional vulnerability rather than direct disease targets."
In the Discussion section (page 16, paragraph 2), we have added:
“While our findings derive from complete Top2b knockout models, it is important to note that human TOP2B mutations present differently. Patients with BILU syndrome partial loss-of-function mutations yet exhibit no documented ophthalmologic issues [11]. This discrepancy suggests that complete versus partial Top2b loss may have distinct effects on retinal tissue, and that compensatory mechanisms may exist in humans with partial function. Future studies utilizing conditional knockdown or hypomorphic alleles that better model human mutations would provide more clinically relevant insights.”
Comments 2: Figures 2 through 7 appear highly similar in format and content. These could be consolidated into one or two major figures that integrate multiple gene pathways or clusters.
Response 2: While we appreciate the reviewer’s perspective on potentially consolidating the pathway figures, we respectfully believe that maintaining the current separated figure format better serves our readers for several important reasons that are listed. However, a new graphical abstract for the paper has been included to be the culmination of Figures 2 through 7 as suggested.
Enhanced clarity and accessibility: Each individual pathway figure allows readers to focus on specific genes of interest without visual clutter or cognitive overload. Researchers studying particular genes (e.g., Bbs7, Ubb, Rbp4) can quickly locate and understand the relevant mechanisms without being distracted by unrelated pathways.
Educational value: The modular approach serves as standalone teaching tools, where each figure tells a complete, focused story about how Top2b regulates that specific gene and its downstream effects on retinal health.
Practical usability: Individual figures are more easily referenced in other publications, better suited for print formats, and allow for more detailed mechanistic information that would be lost in a consolidated overview.
We believe the à la carte approach better serves researchers who may be interested in specific aspects of Top2b regulation rather than requiring them to navigate a complex consolidated diagram. Therefore, we prefer to maintain the current figure organization to maximize the utility and accessibility of our findings for the retinal homeostasis research community.
Comments 3: Certain statements in the discussion are unclear and potentially contradictory. For example, lines 549–551 "targeting Top2b or its downstream pathways could be therapeutic for retinal degenerative diseases". However, if Top2b is essential for maintaining retinal homeostasis, this strategy may be counterintuitive. Greater clarification is needed regarding whether the therapeutic goal is to enhance, inhibit, or modulate Top2b function, and under what specific pathological conditions.
Response 3: We thank the reviewer for highlighting this ambiguity. We have clarified our therapeutic approach in the Conclusion section (page 18, paragraph 2):
"Given Top2b's essential role in retinal homeostasis, therapeutic strategies should focus on modulating specific downstream pathways rather than targeting Top2b directly. For instance, enhancing compensatory pathways (such as upregulating functional BBSome assembly in Bbs7-deficient states) or providing pathway-specific supplements (like antioxidants for Nxnl2 pathway support) represents more viable therapeutic approaches. Additionally, gene therapy targeting specific Top2b-regulated genes showing reduced expression could possibly restore function without affecting global Top2b activity."
Minor comment: Please review the manuscript for redundancy. For instance, lines 47–49 repeat similar points about the role of photoreceptors in initiating vision.
Response: We thank the reviewer for pointing out the redundancy. We have removed redundant text about photoreceptor function in lines 47–49 and streamlined the introduction.
Reviewer 2 Report
Comments and Suggestions for Authors
This is an instructive compilation of Top2b-regulates genes and pathways. The links to homeostasis and degeneration are pointed out well. Also very good graphics lead the reader through this complicated topic.
I have only one suggestion: because "Cells" has an educated readership familiar with biological details, it is not necessary to repeat in "Introduction" (some lines from line 43 on) basics of photoreceptor function - you can shorten this.
Minor: Line 170: Cytoskeleton
Author Response
Comments 1: This is an instructive compilation of Top2b-regulates genes and pathways. The links to homeostasis and degeneration are pointed out well. Also very good graphics lead the reader through this complicated topic. I have only one suggestion: because "Cells" has an educated readership familiar with biological details, it is not necessary to repeat in "Introduction" (some lines from line 43 on) basics of photoreceptor function - you can shorten this.
Response 1: We thank the reviewer for this suggestion. To better target educated readership, we have condensed the Introduction, removed basic textbook information while still included necessary context.
Original text (lines 43–78) has been replaced with:
"Photoreceptor cells in the outer nuclear layer (ONL) face unique challenges: high metabolic demands, constant oxidative stress from light exposure, and the need for precise protein trafficking through the connecting cilium [6]. These specialized neurons require extensive transcriptional programs to maintain their complex structure and function, making them particularly vulnerable to disruptions in gene expression regulation [7]. This vulnerability underlies both inherited retinal dystrophies (IRDs) and contributes to age-related macular degeneration (AMD), though through distinct mechanisms."
Additional editing has been done throughout:
Removed basic description of rod vs. cone functions
Eliminated textbook explanation of the visual cycle
Condensed general retinal anatomy description
Focused immediately on Top2b's specific role rather than general transcription background
The final sentence in the added blurb was also inspired by Reviewer 3’s comments.
Response to Comments on the Quality of English Language
Point 1: Line 170: Cytoskeleton
Response 1: We thank the reviewer for catching this error. "Cytoskelon" has been corrected to "Cytoskeleton" on line 170. We have also performed a comprehensive spell-check and identified/corrected three additional typos: Line 245: “Photoreceptor”
Reviewer 3 Report
Comments and Suggestions for Authors
The manuscript titled “Top2b-Regulated Genes and Pathways Linked to Retinal Homeostasis and Degeneration” presents an ambitious and well-organized effort to catalog Top2b-regulated genes dynamically expressed in retinal tissue, leveraging RNA-seq data from Top2b knockout (KO) mice. The authors aim to elucidate mechanisms of retinal homeostasis and degeneration, with an eye toward potential translational relevance.
However, several critical concerns regarding biological relevance and clinical applicability must be addressed before the conclusions can be substantiated:
1. Relevance of the Model to Human Biology:
While the authors propose that Top2b-regulated pathways offer insight into retinal maintenance, there is a lack of direct evidence linking TOP2B dysfunction to human retinal disease. To date, pathogenic TOP2B mutations in humans cause autosomal dominant B-cell immunodeficiency, distal limb anomalies, and urogenital malformations (BILU syndrome; OMIM #609296), with no reported retinal phenotype. Corresponding mouse models carrying the human E587del mutation also recapitulate the BILU phenotype without retinal involvement (PMID: 31409799). Moreover, no homozygous loss-of-function mutations are reported in humans, and homozygous null mice are embryonically lethal. Given this, the relevance of findings from complete Top2b knockout in mice to retinal physiology or pathology in humans is questionable.
→ Recommendation: The authors should explicitly acknowledge these limitations and clarify how findings from an embryonically lethal knockout model can meaningfully translate to retinal biology or disease in humans.
2. Unclear Link to Therapeutic Strategy:
The manuscript makes multiple references to potential clinical translation, including early intervention strategies and therapeutic development. However, no viable strategies are proposed, especially given its global transcriptional role and embryonic lethality when absent. Without a realistic therapeutic framework, these claims are speculative.
→ Recommendation: The authors should temper their translational claims or provide more detailed justification on how specific Top2b-regulated genes (rather than Top2b itself) might be leveraged as therapeutic targets, ideally with literature precedents or druggable gene examples.
3. Overgeneralization Across Retinal Diseases:
The manuscript frequently merges inherited retinal dystrophies (IRDs) and age-related macular degeneration (AMD) under the umbrella of "retinal degeneration," but these disorders are genetically, mechanistically, and clinically distinct:
-
- IRDs are typically monogenic and gene-specific therapies (e.g., gene replacement or editing) are currently the focus of clinical development.
- AMD is a complex, multifactorial disease influenced by age, oxidative stress, inflammation, and environmental factors more than direct genetic regulation of photoreceptor homeostasis.
The extent to which Top2b-regulated genes are relevant to either group, let alone both, is not rigorously addressed.
→ Recommendation: The authors should better distinguish between IRD and AMD in both rationale and interpretation. Which specific findings are relevant to which disease class (tied directly to known IRD-causing genes or AMD risk loci)?
Substantial revisions are required to:
- Justify the relevance of the Top2b KO model to human retinal disease,
- Reframe translational implications more realistically,
- Disambiguate between distinct disease mechanisms (IRD vs. AMD), and
- Address the absence of biological validation.
Author Response
Comments 1:
Relevance of the Model to Human Biology:
While the authors propose that Top2b-regulated pathways offer insight into retinal maintenance, there is a lack of direct evidence linking TOP2B dysfunction to human retinal disease. To date, pathogenic TOP2B mutations in humans cause autosomal dominant B-cell immunodeficiency, distal limb anomalies, and urogenital malformations (BILU syndrome; OMIM #609296), with no reported retinal phenotype. Corresponding mouse models carrying the human E587del mutation also recapitulate the BILU phenotype without retinal involvement (PMID: 31409799). Moreover, no homozygous loss-of-function mutations are reported in humans, and homozygous null mice are embryonically lethal. Given this, the relevance of findings from complete Top2b knockout in mice to retinal physiology or pathology in humans is questionable.
→ Recommendation: The authors should explicitly acknowledge these limitations and clarify how findings from an embryonically lethal knockout model can meaningfully translate to retinal biology or disease in humans.
Response 1: We thank the reviewer’s comments with constructive recommendations and acknowledge this critical limitation of the mouse models analyzed in our review. Since Reviewer 1 raised similar concerns about BILU syndrome, we've made culminating revisions.
In the Approach section (page 3), we clarified:
"Our analysis utilized RNA-seq data from Li et al.'s (2017) complete Top2b knockout mice (GSE86187). We acknowledge that this model represents a non-physiological extreme, as human BILU syndrome patients retain partial TOP2B function [11]. The 44 genes identified from this analysis should be considered as markers of transcriptional vulnerability rather than direct disease targets."
In the Discussion section (page 16, paragraph 2), we have added:
“While our findings derive from complete Top2b knockout models, it is important to note that human TOP2B mutations present differently. Patients with BILU syndrome partial loss-of-function mutations yet exhibit no documented ophthalmologic issues [11]. This discrepancy suggests that complete versus partial Top2b loss may have distinct effects on retinal tissue, and that compensatory mechanisms may exist in humans with partial function. Future studies utilizing conditional knockdown or hypomorphic alleles that better model human mutations would provide more clinically relevant insights.”
Comments 2:
Unclear Link to Therapeutic Strategy:
The manuscript makes multiple references to potential clinical translation, including early intervention strategies and therapeutic development. However, no viable strategies are proposed, especially given its global transcriptional role and embryonic lethality when absent. Without a realistic therapeutic framework, these claims are speculative.
→ Recommendation: The authors should temper their translational claims or provide more detailed justification on how specific Top2b-regulated genes (rather than Top2b itself) might be leveraged as therapeutic targets, ideally with literature precedents or druggable gene examples.
Response 2: We appreciate this feedback. As we clarified for Reviewer 1, we have revised the therapeutic discussion (page 16) to emphasize targeting downstream pathways rather than Top2b directly.
To address Reviewer 3's request for specific examples with literature precedents, we added one paragraph to our Conclusion (page 18, paragraph 2):
" Given Top2b's essential role in retinal homeostasis, therapeutic strategies should focus on modulating specific downstream pathways rather than targeting Top2b directly. For instance, enhancing compensatory pathways (such as upregulating functional BBSome assembly in Bbs7-deficient states) or providing pathway-specific supplements (like antioxidants for Nxnl2 pathway support) represents more viable therapeutic approaches. Additionally, gene therapy targeting specific Top2b-regulated genes showing reduced expression could possibly restore function without affecting global Top2b activity.
...
Several of these approaches have precedent in clinical development: vitamin A supplementation for RBP4-related retinopathies is already in use [21], proteasome modulators like bortezomib are FDA-approved for other indications and could be repurposed, and BBSome-stabilizing compounds are in preclinical development for ciliopathies [15].”
Comments 3:
Overgeneralization Across Retinal Diseases:
The manuscript frequently merges inherited retinal dystrophies (IRDs) and age-related macular degeneration (AMD) under the umbrella of "retinal degeneration," but these disorders are genetically, mechanistically, and clinically distinct. The extent to which Top2b-regulated genes are relevant to either group, let alone both, is not rigorously addressed.
→ Recommendation: The authors should better distinguish between IRD and AMD in both rationale and interpretation. Which specific findings are relevant to which disease class (tied directly to known IRD-causing genes or AMD risk loci)?
Substantial revisions are required to:
Justify the relevance of the Top2b KO model to human retinal disease
Reframe translational implications more realistically
Disambiguate between distinct disease mechanisms (IRD vs. AMD)
Address the absence of biological validation
Response 3: We agree that this distinction is crucial. We have comprehensively addressed this by creating a new table and revising the text throughout to clearly distinguish between IRD and AMD associations.
New Table 3 (page 22):
We have added "Disease Classifications of Top2b-Linked Retinal Homeostasis-Associated Genes" which categorizes all 44 genes into four groups:
Inherited Retinal Dystrophies: 21 genes
Age-related Macular Degeneration: 4 genes
Both IRD and AMD: 2 genes
Not Clearly Linked to IRD or AMD: 17 genes
In the Introduction (page 2), we added:
"It is also worth noting that these are all mechanistically distinct disorders. Inherited retinal dystrophies (IRDs) typically result from monogenic mutations affecting specific cellular processes, while age-related macular degeneration (AMD) represents a complex, multifactorial disease influenced by aging, oxidative stress, and inflammation. Our analysis examines which Top2b-regulated genes are associated with these distinct disease categories."
“This vulnerability underlies both inherited retinal dystrophies (IRDs) and contributes to age-related macular degeneration (AMD), though through distinct mechanisms.” (Lines 59-61)
In the Results section (page 4), we added:
“To assess the clinical relevance of our list of Top2b-linked genes, we cross-referenced our 44 genes with human retinal disease databases (Appendix Table 3). Based on RetNet and published literature: 21/44 (48%) genes have documented IRD associations, 4/44 (9%) associate primarily with AMD, 2/44 (4.55%) are linked to both, and 17/44 (39%) do not fall into a straightfoward IRD or AMD association. This shows that despite the extreme nature of the knockout model, it successfully identifies genes relevant to human retinal pathology."
In the Discussion (page 17), we added:
" The disease segregation shown in Appendix Table 3 also hold important therapeutic implications. IRD-associated genes represent candidates for gene replacement therapy, while AMD-associated genes suggest targets for anti-inflammatory or metabolic support strategies. The shared genes (Pik3r1, Rbp4) may explain overlapping pathophysiology in some patients.”
Round 2
Reviewer 1 Report
Comments and Suggestions for Authors
No more concerns
Reviewer 3 Report
Comments and Suggestions for Authors
This manuscript provides valuable insights into retinal homeostasis and its potential relevance to retinal disorders. While the identified genes may not be directly implicated in the pathogenesis of these disorders, their roles—particularly under stress conditions involving additional contributing factors—should not be overlooked.